# Towards Realistic Ultrasound Fetal Brain Imaging Synthesis

**Michelle Iskandar**[*2] **Harvey Mannering**[*1] **Zhanxiang Sun**[*1] **Jacqueline Matthew**
[2] **Hamideh Kerdegari** [2] **Laura Peralta**[2] **Miguel Xochicale**[*1] M.XOCHICALE@KCL.AC.UL
[1] *University College London* [2] *King's College London*

## Abstract

Prenatal ultrasound imaging is the first-choice modality to assess fetal health. Medical image datasets for AI and ML methods must be diverse (i.e. diagnoses, diseases, pathologies, scanners, demographics, etc), however there are few public ultrasound fetal imaging datasets due to insufficient amounts of clinical data, patient privacy, rare occurrence of abnormalities in general practice, and limited experts for data collection and validation. To address such data scarcity, we proposed generative adversarial networks (GAN)-based models, diffusion-super-resolution-GAN and transformer-based-GAN, to synthesise images of fetal ultrasound brain planes from one public dataset. We reported that GAN-based methods can generate 256x256 pixel size of fetal ultrasound trans-cerebellum brain image plane with stable training losses, resulting in lower Fréchet inception distance (FID) values for diffusion-super-resolution-GAN (average 7.04 and lower FID 5.09 at epoch 10) than the FID values of transformer-based-GAN (average 36.02 and lower 28.93 at epoch 60). The results of this work illustrate the potential of GAN-based methods to synthesise realistic high-resolution ultrasound images, leading to future work with other fetal brain planes, anatomies, devices and the need of a pool of experts to evaluate synthesised images. Code, data and other resources to reproduce this work are available at https://github.com/budai4medtech/midl2023.

**Keywords:** Medical Image Synthesis, Ultrasound Fetal Imaging, GANs

## 1. Introduction

Prenatal imaging is performed to assess various aspects of pregnancy, including confirmation of the pregnancy, screening for developmental defects, and investigation of pregnancy complications (Kline–Fath and Bitters, 2007). In the last decade, the fields of machine learning (ML) and artificial intelligence (AI) have been successful to model intelligent behaviors with minimal human interference (Hamet and Tremblay, 2017). Particularly, automatic classification of fetal ultrasound planes and fetal head biometric measurement (Burgos-Artizzu et al., 2020b; Sin, 2018; Fiorentino et al., 2022). Despite such advances, there are few challenges faced in prenatal imaging: (a) the accuracy of recorded measurements which can be caused by differences in intra-view variability of imaging equipment and inter-observer variability of sonographer skills (England, 2015; Sarris et al., 2012; Villar et al., 1989; Kesmodel, 2018), (b) availability of expert clinicians or trained technicians to select, to classify and to validate regions of interest (Burgos-Artizzu et al., 2020a), (c) the insufficient and limited amount of clinical data (Jang et al., 2018; Sin, 2018; He et al., 2021), (d) data accessibility due to patient privacy or protection of personal health information (Shin et al., 2018), and (e) the cost of acquisition of clinical data as it requires expensive imaging equipment and experts for data collection and validation (Wang et al., 2019; Kim et al., 2019). Given the advances with generative adversarial networks (GAN) methods to handle

---

[*] Contributed equally

problems in medical reconstructions, image resolution, enhancement, segmentation, lesion detection, data simulation or classification (AlAmir and AlGhamdi, 2022), we hypothesize that realistic ultrasound imaging synthesis can address challenges in data scarcity, accessibility and expensiveness. For instance, Eli et al. (2017) proposed a method of generating freehand ultrasound image simulation using a spatially conditioned GAN. Kazeminia et al. (2020) presented a review of the state-of-the-art research in GAN in medical imaging for classification, denoising, reconstruction, synthesis, registration, and detection. Montero et al. (2021) proposed a method to generate fetal brain US images using an unconditional GAN, StyleGAN2, specifically to improve the fine-grained plane classification, specifically the trans-thalamic and trans-ventricular plane. Hence, the aim of this work is to show the potential of GAN-based methods to generate realistic ultrasound fetal trans-cerebellum brain plane imaging with small datasets.

## 2. Methods and datasets

### 2.1. Diffusion-Super-Resolution-GAN (DSR-GAN)

We use a Denoising Diffusion Probabilistic Model (DDPM) (Ho et al., 2020) due to its recent success in unconditional image synthesis. Computational resources were limited. Therefore to reduce computation time, we finetune a pretrained DDPM to produce 128x128 pixel images. Upscaling to 256x256 using bilinear interpolation yields an FID score of 8.93. To enhance this score, a superresolution model is employed. Both diffusion and GAN-based approaches were explored, but computational limitations led to the selection of Super-Resolution-GAN (SRGAN) (Ledig et al., 2017). The DDPM and SRGAN were trained separately. Histogram matching (Castleman, 1996) is applied after DDPM and before SRGAN to align the synthetic image color distribution with real images. Random zooming, rotating, and horizontal flipping augmentations diversify the dataset.

### 2.2. Transformer-based-GAN (TB-GAN)

The Transformer-based GAN was chosen in order to reproduce longer-distance spatial relationships found in the original images with attention mechanism. This approach aims at generating more coherent images that maintain similar semantic layouts as the original ones. Meanwhile, StyleSwin implements a window attention mechanism that effectively reduces the memory usage in training, enabling synthesize images of higher resolutions (Zhang et al., 2022). Differentiable data augmentation (DiffAug) and adaptive pseudo augmentation (APA) are implemented for StyleSwin because GANs are prone to model collapse and discriminator over-fitting when there are limited data. The two augmentations helped stabilize training for GAN (Zhao et al., 2020; Jiang et al., 2021).

### 2.3. Image Quality Assessment

Quality of synthesised images are evaluated with Fréchet inception distance (FID), measuring the distance between distributions of synthesised and original images (Heusel et al., 2017). The lower the FID number is, the more similar the synthesised images are to the original ones. FID metric showed to work well for fetal head ultrasound images compared to other metrics (Bautista et al., 2022).

## 2.4. Datasets

Trans-cerebellum brain plane ultrasound images from Voluson E6 were used for this work, consisting of 408 training images (Burgos-Artizzu et al., 2020a,b). Scans were collected by multiple operators of similar skill level at BCNatal hospital during standard clinical practice between October 2018 and April 2019. DICOM images were collected and anonymised using png format, resulting in images of various pixel size (e.g., 692x480, 745x559, and 961x663). Note that such datasets only contain healthy participants.

## 3. Experiments: Design and results

Diffusion model was finetuned for 10000 epochs with the Adam optimiser to then train SRGAN for 200 epochs from scratch with the Adam optimiser. The images used to train both models are flipped horizontally, zoomed and rotated randomly to increase the variety of the dataset(Fig 1c). Transfer learning is used when training StyleSwin. The model was firstly pre-trained for 500 epochs on Trans-thalamus plane, which contains more images (1072). Then, the model was fine-tuned on Trans-cerebellum plane images for an additional 200 epochs. Adam optimizer was also used during both pre-training and fine-tuning stages, following the two time-scale update rule with learning rates of 2e-4 for the discriminator and 5e-5 for the generator (van den Heuvel et al., 2018).

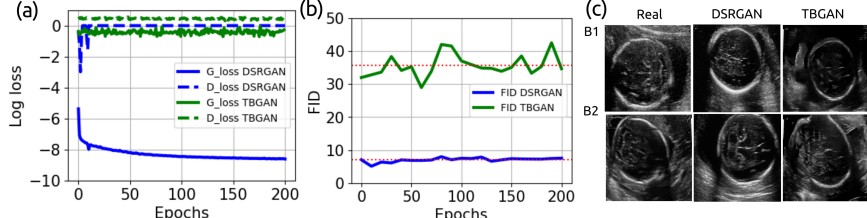

Figure 1: Results from Diffusion-Super-Resolution-GAN (DSR-GAN) and transformer-based-GAN (TB-GAN) models: (a) Convergency of training losses for Generator and Discriminator networks, (b) FID scores: DSR-GAN lower average 7.04 than TB-GAN average 36.02, and (c) 256x256 pixel size trans-cerebellum images of two randomised batches (B1, B2) of real and models.

## 4. Conclusions and future work

Synthesising fetal brain images with the diffusion-Super-Resolution-GAN and transformer-based-GAN methods were successful, generating images of 256x256 pixel size resolution with stable loss values and resulting in lower FID values for Diffusion-Super-Resolution-GAN (average 7.04 and lower 5.09 at epoch 10) compared to FID values of Transformer-based-GAN (average 36.02 and lower 28.93 at epoch 60). The limitations of this work are in the generated 256x256 pixel size image resolutions due limited hardware access and the synthesised images for only healthy participants. However, reported results suggest future work with the potential to synthesise realistic higher-resolution fetal ultrasound images for other anatomies, ultrasound-devices and abnormalities, which can facilitate downstream tasks such as classification or segmentation of fetal ultrasound images.

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
