# OpenReview forum: "Towards Realistic Ultrasound Fetal Brain Imaging Synthesis"
_MIDL.io/2023/Short_Paper_Track — MIDL 2023 Short paper track Poster_

### Official Review · Reviewer_YsSG · 2023-04-11
**Review of "Towards Realistic Ultrasound ..."**

**Rating:** 9
**Confidence:** 4

**Review:**

The aim of this paper is to show the potential of GAN-based methods to generate realistic ultrasound fetal trans-cerebellum brain plane imaging with a small dataset. Namely, 2 models are experienced: a diffusion-super-resolution-GAN and transformer-based-GAN, the first one providing the best result according to Frechet inception distance.

The authors provide their code to reproduce their experiments

The paper is very well written and is technically sound.

Some comments/questions:

For the reader unfamiliar with the image synthesis literature, it is unclear why these specific models were chosen, ie why diffusion + SRGAN, why some specific types of data augmentation (DiffAug, APA) are used for the 2nd model only?

Since the authors mention the « rare occurrence of abnormalities in general practice » in their abstract as an argument to generate synthetic images, they could specify whether their dataset include healthy images or not.

Qualitative results are convincing. DSR-GAN is performing better, in terms of FID, but what the reader wonders is the meaning of an FID value of 7 or 36: is it acceptable for subsequent tasks that would make use of these images? and is training time a critical variable in this case? The comparison of performance is mentioned only in the abstract, maybe this should be in the main text.

The interest of Figure (a) is unclear, especially since the generative loss curves for the 2 models have opposite direction, ie one is increasing and the other one is decreasing. Maybe it is normal, but the reader unfamiliar with loss curve in GAN models may wonder about it.

minor comments or typos
====================
- FID: acronym undefined in the abstract
- hypnotise: seems to be hypothesize?
- anominsied

---

### Official Review · Reviewer_JFQw · 2023-04-21
**Interesting problem, novelty unclear**

**Rating:** 5
**Confidence:** 4

**Review:**

Authors evaluate two generative models to synthesize prenatal US images of fetal brains. Authors find that one of the (GAN) models outperforms the other and is able to generate images that qualitatively share similarities with real images.

Strengths

-	Addressing shortage of data with generative models in medical imaging makes sense.
-	Authors provide code on GitHub.
-	Authors use some interesting combinations of diffusion models and GANs.

Weaknesses
- Synthesizing 256x256 pixel images with GANs is not necessarily state-of-the-art, as current models are able to synthesize much higher resolution images (e.g. StyleGAN). This resolution also falls short of the real data resolution that the authors describe in Sec. 2.4.
- It’s unclear if in the Diffusion-Super-Resolution-GAN (DSR-GAN), both models are jointly trained end-to-end or not.
- There is no real structure in the similarities and differences of the methods evaluated in this paper. Both appear to be off-the-shelf image synthesis models, but it’s unclear why these two were chosen.
- There is no downstream task, and there’s no label or reference segmentation generated for the images. The use case is a bit unclear, what would these images be used for?

Constructive feedback
- Hypnotise --> hypothesize
- 'realistic ultrasound imaging' --> 'realistic ultrasound image synthesis' (?)